# Functional Analysis of Haplotypes in Bovine *PSAP* Gene and Their Relationship with Beef Cattle Production Traits

**DOI:** 10.3390/ani11010049

**Published:** 2020-12-29

**Authors:** Haidong Zhao, Mingli Wu, Xiaohua Yi, Xiaoqin Tang, Pingbo Chen, Shuhui Wang, Xiuzhu Sun

**Affiliations:** 1College of Animal Science and Technology, Northwest A&F University, Yangling 712100, Shaanxi, China; 2018060160@nwafu.edu.cn (H.Z.); wumingli@nwafu.edu.cn (M.W.); yixiaohua@nwafu.edu.cn (X.Y.); txq@nwafu.edu.cn (X.T.); 2017010676@nwafu.edu.cn (P.C.); wangshuhui@nwafu.edu.cn (S.W.); 2College of Grassland Agriculture, Northwest A&F University, Yangling 712100, Shaanxi, China

**Keywords:** *PSAP*, haplotypes, bovine, morphological traits, miR-184, mRNA secondary structure

## Abstract

**Simple Summary:**

With the rapid development of information technology and molecular biotechnology, animal molecular breeding technology is playing an increasingly important role in beef cattle breeding. Prosaposin (*PSAP*) is involved in regulating the growth and development of animals, and it is reported that *PSAP* is an important marker-assisted selection (MAS) in cattle herd. The purpose of this study was to explore the novel variants in 3’ UTR of cattle *PSAP* and evaluate their effects on the morphological traits of four Chinese cattle breeds. In this study, 13 variants were identified in the *PSAP* 3’ UTR from 501 individuals belonging to four cattle breeds. In Nanyang cattle, the distribution of haplotypes was different from the other three breeds. Two groups of haplotypes had association with morphological traits by changing the secondary structures of *PSAP* 3’ UTR rather than the miR-184 target sites. This study not only expands the genetic variation spectrum of cattle *PSAP* but also contributes to MAS genetics and breeding of Chinese cattle breeds.

**Abstract:**

The purpose of this study was to explore functional variants in the prosaposin (*PSAP*) three prime untranslated region (3’ UTR) and clarify the relationship between the variants and morphological traits. Through Sanger sequencing, 13 variations were identified in bovine *PSAP* in four Chinese cattle breeds, with six of them being loci in 3’ UTR. In particular, Nanyang (NY) cattle had a special genotype and haplotype distribution compared to the other three breeds. NY cattle with ACATG and GCGTG haplotypes had higher morphological traits than GTACA and GTACG haplotypes. The results of dual-luciferase reporter assay showed that ACATG and GCGTG haplotypes affected the morphological traits of NY cattle by altering the secondary structure of *PSAP* 3’ UTR rather than the miR-184 target sites. The findings of this study could be an evidence of a complex and varying mechanism between variants and animal morphological traits and could be used to complement candidate genes for molecular breeding.

## 1. Introduction

With continuous economic development and growth of formalized cultivation, beef cattle breeding is in urgent need of strengthening. Traditional breeding methods are slow to produce results, making it difficult to meet demand. Fortunately, over the last decade, bioinformatics has revolutionized livestock breeding. Compared with traditional breeding, molecular breeding has a number of advantages, such as saving time and shortening the generation interval [1,2]. For example, genome-wide association studies (GWAS) have mapped thousands of genetic variants associated with animal development, which is an unprecedented high-resolution genetic characterization of animal breeding [3,4,5,6]. However, growing evidence suggests that successful livestock breeding requires deep understanding of the regulatory mechanism between the genotype and phenotype relationship [7,8,9,10]. The location of these variants can result in phenotype differences, including the level of DNA methylation, transcription factor binding sites, alternative splicing, and protein translation and modification [11,12,13]. China is rich in cattle genetic resources, including Qinchuan (QC), Nanyang (NY), Jiaxian red (JX), and Luxi (LX) cattle. The four breeds originated from the farming plain and have some excellent traits, such as well-developed muscles.

The prosaposin (*PSAP*) sequence is conserved in mammals and widely expressed in different tissues. Bovine *PSAP* is located in chromosome 10, including 15 exons. The alternative splicing of *PSAP* is reported to skip exon 8 (Gln-Asp-Gln) in several mammals, and our previous studies found that the alternative splicing of bovine *PSAP* was consistent with this (findings not published). *PSAP* code two types of proteins with completely different functions. The first is hydrolyzed into four sphingolipid activator proteins: saposins A, B, C, and D [14]. They are indispensable cofactors for the intralysosomal degradation of a number of sphingolipids within the lysosome. The second is secreted in semen, cerebrospinal fluid, milk, and other body fluids. They are involved in the nervous system, collective oxidative stress, sperm fertilization, and other biological processes [15]. *PSAP* is reported to be the target gene of *androgen receptor* (*AR*), and a potential molecular marker was identified in *AR* in our previous studies [16]. At the same time, *PSAP* regulates caspase, mitogen-activated protein kinase (MAPK), phosphatidylinositol 3’-kinase/AKT serine/threonine kinase 1 (PI3K/Akt), and transforming growth factor-β (TGF-β) pathways and has an important relationship with animal growth and development [17]. *PSAP* has been reported as a potential target gene in cattle breeding with functional single-nucleotide polymorphism (SNP) and insertion/deletion (InDels) loci associated with cattle morphological traits [18,19]. Considering their location, we speculate that the cattle *PSAP* gene also has other functional variations.

Therefore, the aim of this study was to explore the novel variants in bovine *PSAP* 3’ UTR and evaluate their effects on morphological traits in four cattle breeds. This study will not only extend the spectrum of genetic variations of cattle *PSAP* but also contribute to marker-assisted selection (MAS) in genetics and breeding in Chinese cattle breeds.

## 2. Materials and Methods

All experimental procedures were performed in accordance with the Regulations for the Administration of Affairs Concerning Experimental Animals approved by the State Council of People’s Republic of China. The study was approved by The Institutional Animal Care and Use Committee of Northwest A&F University (Permit Number: NWAFAC1019).

### 2.1. Samples and Data Collection

In order to explore variations in bovine *PSAP* 3’ UTR, 501 cattle samples from four breeds (LX, n = 104, female; QC, n = 123, female; NY, n = 137, female; JX, n = 137, female) were collected (Table 1). All animals within a breed were managed in the same condition, and sufficient feed was provided by total metabolic rate (TMR). Their morphological traits were measured, including body weight (BW), body height (BH), body length (BL), chest circumference (ChC), chest depth (ChD), chest width (ChW), hucklebone width (HuW), and hip width (HW). Genome DNA was extracted from the leukocytes of venous blood by the phenol chloroform method [16,20]. All DNA samples underwent quality assurance and were uniformly diluted to 50 ng/μL and stored at −80 °C [16].

### 2.2. Primer Design and Variation Genotyping

According to the cattle reference genome (ARS-UCD1.2) from the NCBI database (http://www.ncbi.nlm.nih.gov/), a pair of specific primers was designed by NCBI primer blast for PCR amplification of cattle *PSAP* 3’ UTR (Table 2). The forward primer was located in the 14th intron, and the reverse primer was located in the downstream region of *PSAP*. PCR amplification was performed in 20 μL final volume containing 10 μL 2 × PCR mix, 0.5 μM forward and reverse primers, 50 ng genomic DNA, and ddH_2_O up to 20 μL. The PCR protocol was touchdown PCR as follows: 5 min predegeneration, followed by 10 cycles at 95 °C for 30 s, 60 °C for 30 s (starting at 60 °C and decreasing by 1 °C per cycle), 72 °C for 30 s, 25 cycles at 95 °C for 30 s, annealing temperature for 30 s, 72 °C for 30 s, and finally extending for 10 min. The PCR products of all individuals were sequenced by Sanger sequencing (Sangon, Shanghai, China) to determine the variation types.

### 2.3. Dual-Luciferase Reporter Assay

Human embryonic kidney cell line (HEK293T) was cultured in Dulbecco’s modified Eagle medium (DMEM; Hyclone, Logan, UT, USA) supplemented with 10% fetal bovine serum (BI, Kibbutz Beit Haemek, Israel), 500 U/mL penicillin, and 100 μg/mL streptomycin (Hyclone, Logan, UT, USA). The cell incubator was controlled at 37 °C and 5% of the CO_2_ volume. Four *PSAP* 3’ UTR haplotypes were cloned into psiCHECK2 plasmid by complementary double-stranded annealing, vector double enzyme restriction, and adapter ligation (Table 1). The cells were cotransfected with a mixture of 500 ng recombined reporter vectors and 3 μL miR-184 mimics. After 48 h, the luciferase activity was measured with a dual-luciferase reporter assay system (Promega, Madison, WI, USA). In the luciferase assay, the mimic NC was the negative control. The fold change of luciferase was calculated by comparing each miRNA to NC according to the manufacturer’s instructions. Primers (Table 2) were synthesized by Sangon Biotech (Shanghai, China) Co., Ltd.

### 2.4. Statistical Analysis

Sequences were contrasted and analyzed by Snapgene (GSL Biotech, Chicago, IL, USA). All the population genetic data were calculated and analyzed using the website www.Msrcall.com, including the Hardy–Weinberg equilibrium (HWE), homozygosity (Ho), heterozygosity (He), effective allele number (Ne), and polymorphism information content (PIC). Every cattle breed was collected in the same farm and were of the same gender (female), had a similar age (2–6 years old), had the same feeding management, and had no genetic relationship within a population. The records of morphological traits were analyzed in each cattle breed independently. Mixed linear model analysis was used to establish the influence of different parameters on morphological traits, not including the effects of farm, sex, or age of dam and sire, which had no significant effects on the variation of traits in the four cattle populations in this study. The least squares mean was utilized for morphological traits among the different genotypes and haplotypes: *Yi* = *μ* + *Gi* + *ei*, where *Yi* is the phenotypic value of morphological traits; *μ* is the overall population mean; *Gj* is the genotype; and *ei* is the random error. The chi-square test and ANOVA were used with SPSS software 18.0 (IBM, San Francisco, CA, USA) to test the genotype distribution and the association with morphological traits in different cattle breeds. The least significant difference (LSD) was used for multiple comparison. Statistical significance was measured at *p* < 0.05 and *p* < 0.01. Haplotype and linkage disequilibrium (LD) analyses were done using the website http://analysis.bio-x.cn/myAnalysis.php, and LD heatmap (*R*) and mRNA secondary structure were predicted using the online software RNAfold.

## 3. Results

### 3.1. Thirteen Variations Were Identified in Bovine PSAP

Through DNA sequencing, 13 variations were found in a 167 bp DNA fragment in bovine *PSAP* 3’ UTR, including 12 SNP loci and one deletion locus. Six of the SNPs were located in *PSAP* 3’ UTR. P1 (NC_007329.6g 23314G > A), P2 (NC_007329.6g 23323T > C), and P3 (NC_007329.6g 23324T > C) were located in the 14th intron of *PSAP*. P4 (NC_007329.6g 23354G > A) and P5 (NC_007329.6g 23355T > C) were located in the 15th exon of *PSAP*. The P4 locus was a missense mutation of Arg (AGG) to Lys (AAG), and the P5 locus was a synonymous mutation of Arg (CGT > CGC). P6 (NC_007329.6g 23374G > A), P7 (NC_007379.6g 23314C > T), P8 (NC_007381.6g 23314C > T), P9 (NC_007329.6g 23385A > G), P10 (NC_007329.6g 23386C > T), and P11 (NC_007329.6g 23404A > G) were located in *PSAP* 3’ UTR. P12 (NC_007329.6g 23441G > A) and P13 (NC_007329.6g 23473 del “T”) were located in the downstream region of *PSAP* (Figure 1). P6–11 were named N1, N6, N8, N12, N13, and N31, respectively, to indicate the location of the mutation in 3’ UTR.

### 3.2. Population Parameters of 13 Variations in Four Cattle Breeds

The population parameters of four cattle breeds were calculated, including genotype frequency, allele frequency, HWE, Ho, He, Ne, and PIC (Table 3 and Table 4). For the P3 locus, CC was the dominant genotype in LX, NY, and JX cattle, but TT was the dominant genotype in QC cattle (Table 3). Chi-square test (Figure 2) showed the difference in allele and genotype frequency between QC cattle and the other three breeds more clearly. In the P8 locus, TT was the dominant genotype in NY, while CC was the dominant genotype in the other three groups (Table 4). Figure 2 more clearly shows the difference in genotype and allele frequency between NY and the other three breeds at locus P8. Incidentally, for the N6 (P7) locus, the quantity of TT and TC genotypes were not enough for statistical calculation in the present study (Table 4). The rest of the genotype distribution of the 11 loci were the same in the four cattle breeds (Table 3 and Table 4). Linkage disequilibrium analysis showed that the linkage of 13 loci in QC, LX, and JX cattle was different from the NY cattle (Figure 3).

### 3.3. Association between 13 Variations and Morphological Traits

The associations between the 13 novel variants and cattle morphological traits were investigated. Seven of them were associated with morphological traits: P1, P3, P4, N1, N12, N31, and P12. For the P1 locus, JX cattle with AA genotype had higher waist width and hucklebone width than GG genotype, but NY cattle with GG genotype had higher body length than AA genotype (*p* < 0.05). For the P3 locus, LX cattle with CC genotype had higher abdominal circumference than TT genotype, and the TC genotype was the best genotype (*p* < 0.05). For the P4 locus, NY cattle with AA genotype had higher waist width than GG genotype, but LX cattle with GG genotype had higher abdominal circumference than AA genotype, and the GA genotype was the best genotype (*p* < 0.05). For the N1 locus, NY cattle with GG and AA genotypes had higher abdominal circumference than GA genotype (*p* < 0.05). For the N12 locus, JX cattle with AA genotype had higher hucklebone width than GA genotype (*p* < 0.05). For the N31 locus, NY cattle with AA and GG genotypes had higher abdominal circumference than GA genotype, while JX cattle with AA and GG genotypes had higher body length, waist width, and hucklebone width than GA genotype (*p* < 0.05). For the P12 locus, NY cattle with GG genotype had higher abdominal circumference than CC genotype, and LX cattle with GG genotype had higher waist width and body weight than AA genotype (*p* < 0.05) (Table 5).

### 3.4. Association of Four Haplotypes of PSAP 3’ UTR and Morphological Traits in NY Cattle

Considering the low frequency of the N6 locus, haplotypes of *PSAP* 3’ UTR in four cattle breeds were structured with five SNP loci (N1, N8, N12, N13, and N31). For the N8 locus, the number of allele T and genotype TT were different between NY cattle and the other cattle breeds. Based on the differences in genotype and allele, the haplotype distribution was different between NY cattle and the other cattle breeds. The dominant haplotypes of NY cattle were GTACG, GTACA, CCGTG, and ACATG, but the dominant haplotypes of JX, LX, and QC cattle were GCACG, GCACA, ATATG, and GTGTG (Figure 4).

The relationship between the haplotypes and morphological traits were analyzed in the four cattle breeds. Interestingly, the association between haplotypes and morphological traits only appeared in the NY cattle population. NY cattle with haplotypes ACATG and GCGTG had higher body height, body length, chest circumference, and body weight than those with haplotypes GTACA and GTACG (Table 6).

3’ UTR is an important regulatory region of mRNA stability, and miRNAs widely participate in the regulation process. Based on the miRBase database, bta-miR-184 was found to have different binding capacity with the four haplotypes in NY cattle. Four *PSAP* haplotypes were cloned into psiCHECK2 plasmid. miR-184 mimics, mimics NC, and four haplotypes were cotransfected in the HEK293-T cell line. In the luciferase assay, the relative luciferase of the mimics NC group was higher than the miR-184 mimics group, and the miR-184 did not affect the targeting of *PSAP* 3’ UTR. More interestingly, the relative luciferase between the two groups was different, with the relative luciferase of the ACATG and GCGTG group being higher than the GTACA and GTACG group (Figure 5). Based on the evidence that SNPs could affect the mRNA secondary structure and protein folding, we concluded that two groups of haplotypes in *PSAP* 3’ UTR could contribute to mRNA stability. Through a mRNA secondary structure prediction of four haplotypes by RNAfold, we found that the free energy of the ACATG and GCGTG group was higher than that of the GTACA and GTACG group (Figure 6). This could be the reason for the differences in relative luciferase between the two groups.

## 4. Discussion

It is more strategically important than ever to preserve as much livestock diversity as possible to ensure enough basic gene reserve for future breeding needs [21]. There is less use of beef cattle genetic resources in the beef industry, and most of them are protected by the government [22,23]. Irrespective of the size of the farm, the large majority of breeds are commercial breeds rather than indigenous breeds [24]. Based on the complexity of China’s geography, there is an extremely large amount of genetic resources that could be discovered and utilized from numerous cattle breeds. Molecular breeding is an appropriate choice, and analyzing inherent mechanism is necessary for their popularization and application [25].

*PSAP* has an important function in the Caspase, MAPK, PI3K/Akt, and TGF-β pathways, which have potential function in animal breeding [17]. For example, Guo found six SNPs in the 10th intron and 11th exon of *PSAP* that influenced carcass and meat quality traits [18]. In this study, 13 variations belonging to the 14th intron, 15th exon, 3’ UTR, and downstream region of *PSAP* were found in four cattle breeds. After association analysis, we consider that the P4 locus (missense mutation) and six SNPs in 3’ UTR could be seen as functional variants in beef breeding; however, there was not enough evidence to suggest this. The differences in genotype and allele distribution between NY cattle and the other three cattle breeds attracted our attention. The association between the haplotypes and body height, body length, chest circumference, and body weight were investigated, and we found the morphological traits with the ACATG and GCGTG group was higher than that with the GTACA and GTACG group. This could be regarded as an important basis for a potentially valuable quantitative trait locus (QTL).

The function of variants depends on their location. In the upstream, it can affect DNA methylation, the location of the CpG island, and transcription element recognition sites [26]. In the exon, it can affect protein translation and the spatial structure of proteins. In the intron, it can participate in the function of unknown coding genes, noncoding RNA, and gene alternative splicing. In the 3’ UTR, it can affect targeting with miRNA. miRNA is a kind of small RNA that can target the gene 3’ UTR and construct RNA-induced silencing complex (RISC) [27]. Many studies have demonstrated it as an important mechanism of variants. SNP–miRNA–mRNA interaction networks have an important role in human health and animal production [28,29]. Besides, all the variants in mRNA can affect stability due to the mRNA secondary structure and free energy [30,31]. However, the functional mechanisms underlying the associations are largely unknown [32,33,34]. To more clearly and comprehensively reveal the complex connections between the four haplotypes and miR-184, luciferase assay was performed to investigate the influence of the functional SNP loci. It is a pity that targeting miR-184 was not a mechanism of the functional SNPs in NY cattle *PSAP* 3’ UTR. Compared with the expression of miRNA in muscle and adipose, its expression was not sufficient to mediate such a large effect on the morphological traits in NY cattle [34,35]. However, we found an unexpected phenomenon that the luciferase activity of the ACATG and GCGTG group was higher than the GTACA and GTACG group. To our surprise, the free energy of the ACATG and GCGTG group was lower than the GTACA and GTACG group. We hypothesize that the free energy of the haplotypes in the two groups is the reason for the differences in morphological traits in NY cattle.

More and more novel functional variants are being confirmed by the development of bioinformatics, but most of them lack underlying molecular mechanisms [36,37]. It is an important but difficult work to identify their potential value in animal breeding. Briefly, we identified 13 variations of loci in bovine *PSAP*, including six SNP loci in 3’ UTR. The distribution of haplotypes in NY cattle was different from the other three breeds. Two groups of haplotypes had association with morphological traits by changing the secondary structures of *PSAP* 3’ UTR rather than the miR-184 target sites. Our findings could be evidence of a complex and varying mechanism between variants and animal morphological traits and might be a useful MAS for beef cattle breeds.

## 5. Conclusions

In this study, 13 variants were identified in *PSAP* 3′ UTR from 501 individuals of four Chinese cattle breeds, including three loci in the 14th intron, two loci in the 14th exon, six loci in the 3’ UTR, and two loci in the downstream region of *PSAP*. NY cattle had a particular haplotype distribution compared to the other three cattle breeds. The results of the association analysis showed that NY cattle with haplotypes ACATG and GCGTG had better production traits than those with haplotypes GTACA and GTACG by changing the secondary structures of *PSAP* 3’ UTR rather than the miR-184 target sites.

## Figures and Tables

**Figure 1 animals-11-00049-f001:**
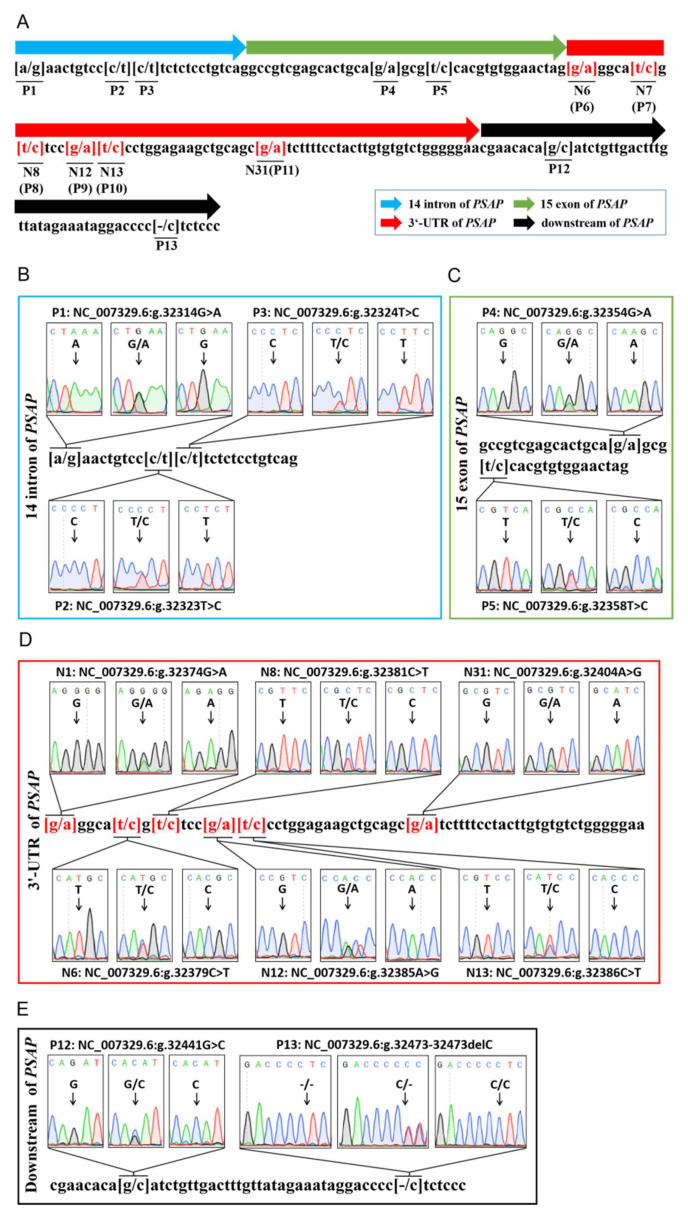
Sequencing of variants in bovine *PSAP*. (**A**) The genetic mapping of bovine *PSAP* by sequencing; (**B**) three single-nucleotide polymorphisms (SNPs) in the 14th intron of bovine *PSAP*; (**C**) two SNPs in the 15th exon of bovine *PSAP*; *(***D**) five SNPs in bovine *PSAP* 3’ UTR; (**E**) two variants in the downstream region of bovine *PSAP*. P: the number variants in the fragment of *PSAP*; N: the number of bases in *PSAP* 3’ UTR.

**Figure 2 animals-11-00049-f002:**
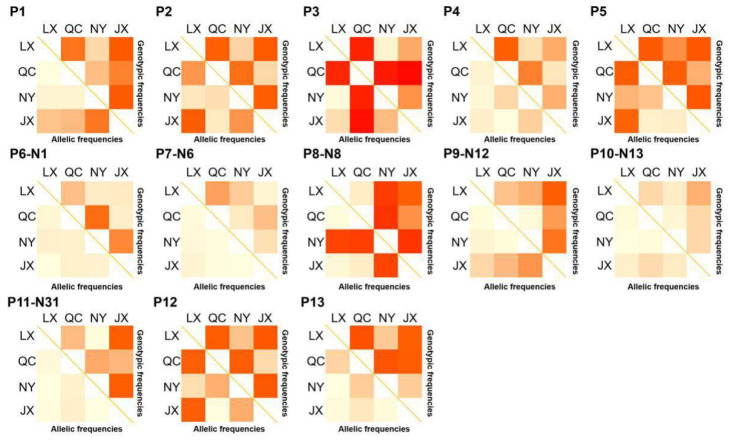
χ^2^ test on genotypic and allele frequency distribution of variations in four Chinese cattle *PSAP*. QC: Qinchuan cattle; NY: Nanyang cattle; JX: Jiaxian red cattle; LX: Luxi cattle.

**Figure 3 animals-11-00049-f003:**
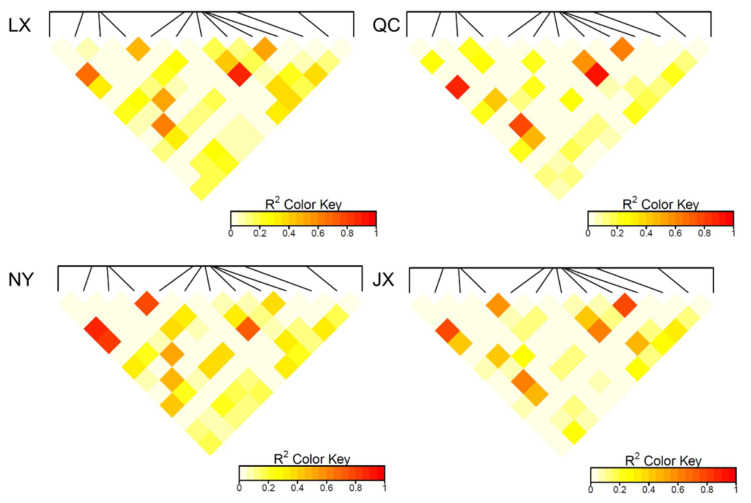
Linkage disequilibrium analysis (*R*^2^) of 13 variants in four cattle breeds. QC: Qinchuan cattle; NY: Nanyang cattle; JX: Jiaxian red cattle; LX: Luxi cattle.

**Figure 4 animals-11-00049-f004:**
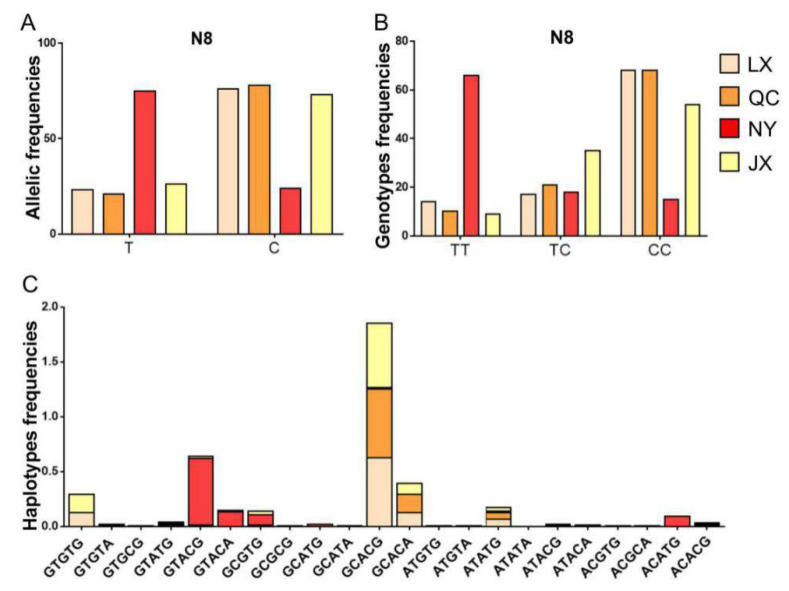
Allele, genotype, and haplotype distribution in four cattle breeds. (**A**) The allele frequency of the N8 locus of bovine *PSAP* in four cattle breeds; (**B**) the genotype frequency of the N8 locus of bovine *PSAP* in four cattle breeds; (**C**) the haplotype frequency of five SNPs of bovine *PSAP* 3’ UTR in four cattle breeds. QC: Qinchuan cattle; NY: Nanyang cattle; JX: Jiaxian red cattle; LX: Luxi cattle.

**Figure 5 animals-11-00049-f005:**
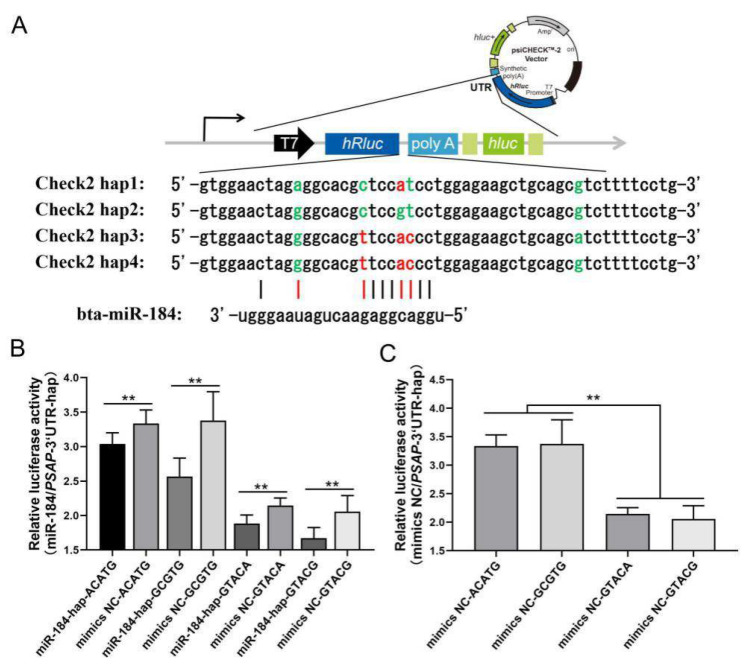
Targeting of bta-miR-184 and four haplotypes of bovine *PSAP* 3’ UTR. (**A**) Four haplotypes and their targeting of bta-miR184 (red represents mismatch and green represents match); (**B**) relative luciferase activity of four haplotypes and bta-miR-184; (**C**) relative luciferase activity of two haplotype groups. ** *p* < 0.01.

**Figure 6 animals-11-00049-f006:**
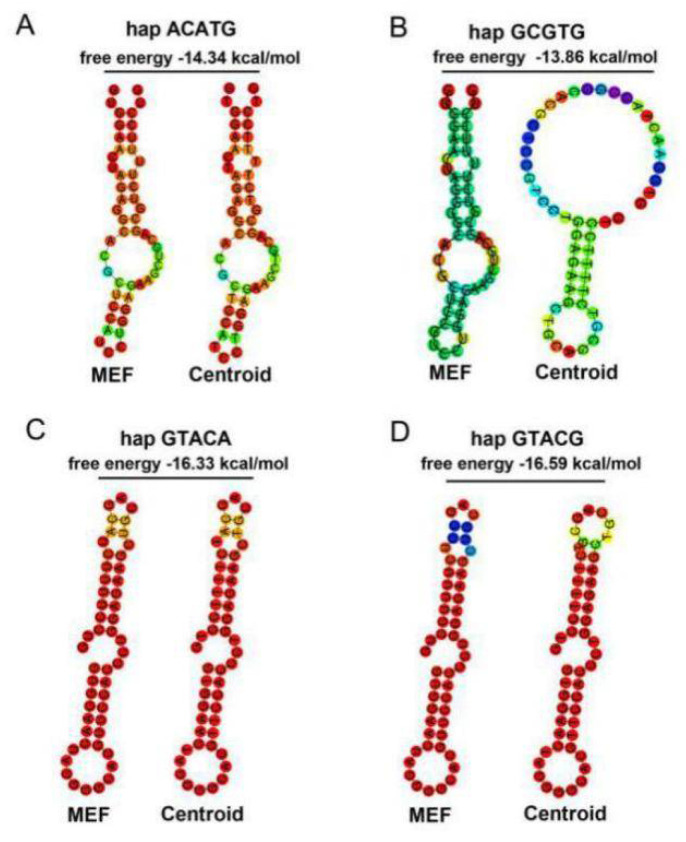
mRNA secondary structure prediction of four haplotypes in *PSAP* 3’ UTR: (**A**) ACATG; (**B**) GCGTG; (**C**) GTACA; (**D**) GTACG.

**Table 1 animals-11-00049-t001:** Information on samples used in this study.

Breed	Sampling Location	Population
QC (Qinchuan cattle)	Yangling, Shaanxi	123
NY (Nanyang cattle)	Nanyang, Henan	137
JX (Jiaxian red cattle)	Nanyang, Henan	137
LX (Luxi cattle)	Jining, Shandong	104

**Table 2 animals-11-00049-t002:** Primers used in the sequencing and vector construction.

Primers	Sequence	Notes
*PSAP*-HAP1-F1	TCGAGgtggaactagaggcacgctccatcctggagaagctgcagcgtcttttcctgGC	Vector construction
*PSAP*-HAP1-R1	GGCCGCcaggaaaagacgctgcagcttctccaggatggagcgtgcctctagttccacC	
*PSAP*-HAP2-F2	TCGAGgtggaactaggggcacgctccgtcctggagaagctgcagcgtcttttcctgGC	Vector construction
*PSAP*-HAP2-R2	GGCCGCcaggaaaagacgctgcagcttctccaggacggagcgtgcccctagttccacC	
*PSAP*-HAP3-F3	TCGAGgtggaactaggggcacgttccaccctggagaagctgcagcatcttttcctgGC	Vector construction
*PSAP*-HAP3-R3	GGCCGCcaggaaaagatgctgcagcttctccagggtggaacgtgcccctagttccacC	
*PSAP*-HAP4-F4	TCGAGgtggaactaggggcacgttccaccctggagaagctgcagcgtcttttcctgGC	Vector construction
*PSAP*-HAP4-R4	GGCCGCcaggaaaagacgctgcagcttctccagggtggaacgtgcccctagttccacC	
*PSAP-F*	GGTGTCGGGTCCTCTTTCTG	Variations screening
*PSAP-R*	GCGTGTCGGCATCTGTCTAG	

*PSAP*: prosaposin; HAP: haplotype.

**Table 3 animals-11-00049-t003:** Genotypes, alleles, He, Ne, and PIC for variations of the 14th intron, 15th exon, and downstream region in cattle *PSAP.*

Locus	Breed	Size	Genotype Frequency			Allele Frequency		HWE	Population Parameters			
		**N**	**GG**	**GA**	**AA**	**G**	**A**	***p*** **Value**	**Ho**	**He**	**Ne**	**PIC**
**P1**	**LX**	104	11	6	87	0.135	0.865	*p <* 0.05	0.767	0.233	1.304	0.206
	**QC**	123	6	21	96	0.134	0.866	*p <* 0.05	0.768	0.232	1.303	0.205
	**NY**	137	8	12	117	0.102	0.898	*p <* 0.05	0.817	0.183	1.225	0.167
	**JX**	137	12	40	85	0.234	0.766	*p <* 0.05	0.642	0.358	1.558	0.294
		**N**	**CC**	**TC**	**TT**	**C**	**T**	***p*** **Value**	**Ho**	**He**	**Ne**	**PIC**
**P2**	**LX**	104	65	19	20	0.716	0.284	*p <* 0.05	0.594	0.406	1.685	0.324
	**QC**	123	93	24	6	0.854	0.146	*p <* 0.05	0.750	0.250	1.333	0.219
	**NY**	137	99	17	21	0.785	0.215	*p <* 0.05	0.662	0.338	1.510	0.281
	**JX**	137	113	21	3	0.901	0.099	*p <* 0.05	0.822	0.178	1.216	0.162
		**N**	**CC**	**TC**	**TT**	**C**	**T**	***p*** **Value**	**Ho**	**He**	**Ne**	**PIC**
**P3**	**LX**	104	88	7	9	0.880	0.120	*p <* 0.05	0.789	0.211	1.268	0.189
	**QC**	123	14	6	103	0.138	0.862	*p <* 0.05	0.762	0.238	1.313	0.210
	**NY**	137	111	13	13	0.858	0.142	*p <* 0.05	0.756	0.244	1.323	0.214
	**JX**	137	122	12	3	0.934	0.066	*p <* 0.05	0.877	0.123	1.140	0.115
		**N**	**GG**	**GA**	**AA**	**G**	**A**	***p*** **Value**	**Ho**	**He**	**Ne**	**PIC**
**P4**	**LX**	104	91	5	8	0.899	0.101	*p <* 0.05	0.818	0.182	1.222	0.165
	**QC**	123	107	15	1	0.930	0.070	*p <* 0.05	0.871	0.129	1.148	0.120
	**NY**	137	113	13	11	0.872	0.128	*p <* 0.05	0.777	0.223	1.287	0.198
	**JX**	137	122	12	3	0.934	0.066	*p <* 0.05	0.877	0.123	1.140	0.115
		**N**	**CC**	**TC**	**TT**	**C**	**T**	***p*** **Value**	**Ho**	**He**	**Ne**	**PIC**
**P5**	**LX**	104	33	21	50	0.418	0.582	*p <* 0.05	0.513	0.487	1.948	0.368
	**QC**	123	9	27	87	0.183	0.817	*p <* 0.05	0.701	0.299	1.426	0.254
	**NY**	137	30	18	89	0.285	0.715	*p <* 0.05	0.593	0.407	1.687	0.324
	**JX**	137	8	47	82	0.230	0.770	*p <* 0.05	0.646	0.354	1.548	0.291
		**N**	**GG**	**GC**	**CC**	**G**	**C**	***p*** **Value**	**Ho**	**He**	**Ne**	**PIC**
**P12**	**LX**	104	48	28	28	0.596	0.404	*p <* 0.05	0.518	0.482	1.929	0.366
	**QC**	123	24	42	57	0.366	0.634	*p <* 0.05	0.536	0.464	1.866	0.356
	**NY**	137	54	30	53	0.504	0.496	*p <* 0.05	0.500	0.500	2.000	0.375
	**JX**	137	21	59	57	0.369	0.631	*p >* 0.05	0.535	0.465	1.871	0.357
		**N**	**CC**	**C-**	**--**	**C**	**-**	***p*** **Value**	**Ho**	**He**	**Ne**	**PIC**
**P13**	**LX**	104	32	14	58	0.375	0.625	*p < 0.05*	0.531	0.496	1.882	0.359
	**QC**	123	26	67	30	0.484	0.516	*p > 0.05*	0.501	0.499	1.998	0.375
	**NY**	137	36	31	70	0.376	0.624	*p < 0.05*	0.531	0.469	1.884	0.359
	**JX**	137	34	44	59	0.409	0.591	*p < 0.05*	0.517	0.483	1.936	0.367

QC: Qinchuan cattle; NY: Nanyang cattle; JX: Jiaxian red cattle; LX: Luxi cattle; HWE: Hardy–Weinberg equilibrium; Ho: homozygosity; He: heterozygosity; Ne: effective allele number; PIC: polymorphism information content.

**Table 4 animals-11-00049-t004:** Genotypes, alleles, He, Ne, and PIC for variations of the cattle *PSAP* 3’ UTR.

Locus	Breed	Size	Genotype Frequency			Allele Frequency		HWE	Population Parameters			
		**N**	**GG**	**GA**	**AA**	**G**	**A**	***p*** **Value**	**Ho**	**He**	**Ne**	**PIC**
**P6 (N1)**	**LX**	104	91	8	5	0.913	0.087	*p* < 0.05	0.842	0.158	1.188	0.146
	**QC**	123	109	13	1	0.939	0.061	*p* > 0.05	0.885	0.115	1.129	0.108
	**NY**	137	115	10	12	0.876	0.124	*p* < 0.05	0.783	0.217	1.278	0.194
	**JX**	137	122	12	3	0.908	0.092	*p* < 0.05	0.877	0.123	1.140	0.115
		**N**	**CC**	**TC**	**TT**	**C**	**T**	***p*** **Value**	**Ho**	**He**	**Ne**	**PIC**
**P7 (N6)**	**LX**	104	101	1	2	0.976	0.024	*p* < 0.05	0.953	0.046	1.049	0.046
	**QC**	123	113	9	1	0.955	0.045	*p* > 0.05	0.914	0.085	1.093	0.082
	**NY**	137	127	7	3	0.953	0.047	*p* < 0.05	0.910	0.090	1.100	0.086
	**JX**	137	132	3	2	0.947	0.053	*p* < 0.05	0.950	0.050	1.052	0.049
		**N**	**TT**	**TC**	**CC**	**T**	**C**	***p*** **Value**	**Ho**	**He**	**Ne**	**PIC**
**P8 (N8)**	**LX**	104	15	18	71	0.231	0.769	*p* < 0.05	0.645	0.355	1.550	0.292
	**QC**	123	13	26	84	0.211	0.789	*p* < 0.05	0.667	0.333	1.500	0.279
	**NY**	137	91	25	21	0.755	0.245	*p* < 0.05	0.630	0.369	1.586	0.301
	**JX**	137	13	49	75	0.266	0.734	*p* > 0.05	0.602	0.398	1.660	0.319
		**N**	**AA**	**GA**	**GG**	**A**	**G**	***p*** **Value**	**Ho**	**He**	**Ne**	**PIC**
**P9 (N12)**	**LX**	104	85	7	12	0.851	0.149	*p* < 0.05	0.746	0.254	1.340	0.221
	**QC**	123	99	16	8	0.870	0.130	*p* < 0.05	0.774	0.226	1.293	0.201
	**NY**	137	113	17	7	0.887	0.113	*p* < 0.05	0.799	0.201	1.251	0.181
	**JX**	137	92	33	12	0.770	0.230	*p* < 0.05	0.670	0.330	1.491	0.275
		**N**	**CC**	**TC**	**TT**	**C**	**T**	***p*** **Value**	**Ho**	**He**	**Ne**	**PIC**
**P10 (N13)**	**LX**	104	72	17	15	0.774	0.226	*P* < 0.05	0.650	0.350	1.538	0.289
	**QC**	123	85	27	11	0.801	0.199	*p* < 0.05	0.681	0.319	1.468	0.268
	**NY**	137	92	29	16	0.777	0.223	*p* < 0.05	0.654	0.346	1.529	0.286
	**JX**	137	82	39	16	0.720	0.280	*p* < 0.05	0.616	0.384	1.623	0.310
		**N**	**AA**	**GA**	**GG**	**A**	**G**	***p*** **Value**	**Ho**	**He**	**Ne**	**PIC**
**P11 (N31)**	**LX**	104	11	7	86	0.139	0.861	*p* < 0.05	0.760	0.240	1.316	0.211
	**QC**	123	11	19	93	0.167	0.833	*p* < 0.05	0.722	0.278	1.385	0.239
	**NY**	137	13	9	115	0.128	0.872	*p* < 0.05	0.777	0.223	1.287	0.198
	**JX**	137	4	27	106	0.124	0.876	*p* > 0.05	0.777	0.223	1.287	0.198

P: the number variants in the fragment of *PSAP*; N: the number of bases in *PSAP* 3’ UTR; QC: Qinchuan cattle; NY: Nanyang cattle; JX: Jiaxian red cattle; LX: Luxi cattle; HWE: Hardy–Weinberg equilibrium; Ho: homozygosity; He: heterozygosity; Ne: effective allele number; PIC: polymorphism information content.

**Table 5 animals-11-00049-t005:** Relationship between variations of *PSAP* and morphological traits.

Locus	Breed	Morphological Trait	Observed Genotypes (Mean ^a^ ± SE)			
			GG	GA	AA	*p* Value
**P1**	**JX**	Waist width (cm)	41.38 ^b^ ± 2.60 (*n* = 8)	42.13 ^ab^ ± 1.43 (*n* = 24)	45.58 ^a^ ± 0.61 (*n* = 69)	0.015
**P1**	**JX**	Hucklebone width (cm)	24.17 ^b^ ± 2.08 (*n* = 12)	23.74 ^b^ ± 1.04 (*n* = 39)	27.91 ^a^ ± 0.50 (*n* = 85)	0.000
**P1**	**NY**	Body length (cm)	146.38 ^a^ ± 5.63 (*n* = 8)	145.08 ^a^ ± 5.41 (*n* = 12)	138.13 ^b^ ± 1.00 (*n* =115)	0.038
			**TT**	**TC**	**CC**	***p*** **Value**
**P3**	**LX**	Abdominal circumference (cm)	183.78 ^c^ ± 5.89 (*n* = 9)	214.71 ^a^ ± 12.81 (*n* = 7)	198.06 ^b^ ± 1.69 (*n* = 88)	0.003
			**GG**	**GA**	**AA**	***p*** **Value**
**P4**	**NY**	Waist width (cm)	44.59 ^b^ ± 0.84 (*n* = 45)	49.83 ^ab^ ± 0.17 (*n* = 3)	51.83 ^a^ ± 5.09 (*n* = 3)	0.045
**P4**	**LX**	Abdominal circumference (cm)	197.79 ^b^ ± 1.68 (*n* = 89)	216.14 ^a^ ± 12.43 (*n* = 7)	183.75 ^c^ ± 6.62 (*n* = 8)	0.002
			**GG**	**GA**	**AA**	***p*** **Value**
**N1**	**NY**	Abdominal circumference (cm)	174.11 ^a^ ± 1.09 (*n* = 114)	160.90 ^b^ ± 6.72 (*n* = 10)	177.77 ^a^ ± 6.95 (*n* = 11)	0.009
			**GG**	**GA**	**AA**	***p*** **Value**
**N12**	**JX**	Hucklebone width (cm)	24.08 ^b^ ± 6.36 (*n* = 12)	24.72 ^ab^ ± 6.61 (*n* = 32)	27.27 ^a^ ± 5.18 (*n* = 92)	0.033
			**GG**	**GA**	**AA**	***p*** **Value**
**N31**	**NY**	Abdominal circumference (cm)	219.65 ^a^ ± 3.80 (*n* = 42)	175.67 ^b^ ± 12.39 (*n* = 3)	211.17 ^a^ ± 6.16 (*n* = 6)	0.011
**N31**	**JX**	Body length (cm)	145.83 ^a^ ± 0.89 (*n* = 106)	139.27 ^b^ ± 1.76 (*n* = 26)	142.00 ^ab^ ± 5.40 (*n* = 4)	0.005
**N31**	**JX**	Waist width (cm)	45.45 ^a^ ± 0.50 (*n* = 83)	38.00 ^b^ ± 2.29 (*n* = 15)	48.33 ^a^ ± 2.33 (*n* = 3)	0.000
**N31**	**JX**	Hucklebone width (cm)	27.28 ^a^ ± 0.49 (*n* = 106)	22.65 ^b^ ± 1.38 (*n* = 26)	27.00 ^ab^ ± 2.65 (*n* = 4)	0.001
			**CC**	**GC**	**GG**	***p*** **Value**
**P12**	**NY**	Abdominal circumference (cm)	228.03 ^a^ ± 3.51 (*n* = 18)	217.00 ^ab^ ± 8.18 (*n* = 13)	204.70 ^b^ ± 5.89 (*n* = 20)	0.016
**P12**	**LX**	Waist width (cm)	48.11 ^a^ ± 0.80 (*n* = 28)	45.50 ^ab^ ± 1.14 (*n* = 28)	44.23 ^b^ ± 0.78 (*n* = 48)	0.011
**P12**	**LX**	Body weight (kg)	426.57 ^a^ ± 12.39 (*n* = 28)	409.04 ^ab^ ± 20.17 (*n* = 28)	367.69 ^b^ ± 10.27 (*n* = 48)	0.029

P: the number variants in the fragment of *PSAP*; N: the number of bases in *PSAP* 3’ UTR; QC: Qinchuan cattle; NY: Nanyang cattle; JX: Jiaxian red cattle; LX: Luxi cattle; a,b,c = *p* < 0.05.

**Table 6 animals-11-00049-t006:** Relationship between four haplotypes of *PSAP* 3’ UTR and morphological traits in NY cattle.

Morphological Trait	Observed Genotypes (LSM^a^ ± SE)				
	AA-CC-AA-TT-GG	GG-CC-GG-TT-GG	GG-TT-AA-CC-AA	GG-TT-AA-CC-GG	*p* Value
**Body height** **(cm)**	134.28 ± 1.34 ^a^ (*n* = 7)	137.00 ± 1.96 ^a^ (*n* = 4)	129.58 ± 1.40 ^b^ (*n* = 12)	128.02 ± 0.59 ^b^ (*n* = 66)	0.001
**Body length** **(cm)**	146.00 ± 2.85 ^a^ (*n* = 7)	152.00 ± 3.14 ^a^ (*n* = 4)	140.42 ± 1.52 ^b^ (*n* = 12)	139.11 ± 1.07 ^b^ (*n* = 66)	0.007
**Chest circumference** **(cm)**	184.57 ± 2.72 ^a^ (*n* = 7)	189.50 ± 2.60 ^a^ (*n* = 4)	176.17 ± 2.62 ^b^ (*n* = 12)	174.16 ± 1.07 ^b^ (*n* = 66)	0.001
**Body weight** **(kg)**	440.86 ± 11.53 ^a^ (*n* = 7)	440.25 ± 5.56 ^a^ (*n* = 4)	398.42 ± 11.41 ^b^ (*n* = 12)	395.06 ± 6.30 ^b^ (*n* = 66)	0.040

LSM: least squares mean; SE: standard error; a,b = *p* < 0.05.

## Data Availability

Data sharing not applicable. No new data were created or analyzed in this study. Data sharing is not applicable to this article.

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
