# Peer review of "Functional Analysis of Haplotypes in Bovine PSAP Gene and Their Relationship with Beef Cattle Production Traits"

_animals, 2020, doi:10.3390/ani11010049_

Round 1

Reviewer 1 Report

The introduction and discussion require moderate editing of the English language. The methods section and results do however read well. For this reason, I have attached 1/2 a page of suggested alterations to the introduction which may demonstrate the level of alterations required. 

This is a very interesting field of research. Identifying advantageous genetic traits in cattle is incredibly important. Therefore, I enjoyed reading the paper, however, the completion of editing will ensure it can be read by all as I struggled at times to understand what was being explained.

Author Response

Thanks for your comments, we have carefully revised our manuscript according to the reviewer's comments. See the annex for details.

Reviewer 2 Report

Authors in the manuscript entitled “Exploring High-density Variants of Bovine PSAP and Their Relationship with beef production” investigated the variants within 3’UTR regions of PSAP gene using 501 individuals from 4 Chinese cattle breeds. Authors identified 30 variants and also explore their association with body measurements and body weight in beef.

In general, It is a typical association paper with not much novelty and creativity. However, authors explored the variants in four different local breeds which its results could be beneficial in future population genetics and association studies.

Major comments:

A similar paper from the same lab has already been published in Animal Biotechnology, https://doi.org/10.1080/10495398.2020.1758122. In that paper authors studied the InDel variants in bovine PSAP gene and their relationship with growth traits using eight cattle breeds. I’m wondering why authors did not presented the results of this study in that paper. Moreover, authors did not cite that paper in this manuscript.

I don’t think body measurement traits are primary traits for beef production industry. If authors include more important economical traits such as carcass, meat quality and body weight traits, then I would say the identified variants could be used in MAS.

Lack of information on statistical models for association analysis. For example: the statistical model, explanatory variables, multiple testing correction.

Minor Comments:

Title: Title of manuscript does not highlight the content of manuscript. Please revise the title. Content of paper should refers to body measurements traits rather than beef production and body weight traits. 30 variants within 3'UTR of PSAP can not be considered as High-density Variants !

Abstract: The abstract put more focus on the introduction than results and conclusion. I would suggest authors revise the abstract completely.

Introduction:

Authors should discuss the results of previous studies on the association of variants within PSAP gene with growth traits. Then, they should report the hypothesis and necessity of current study.

More description about the evolution, distributions and characteristics of four studied cattle breeds are needed.

Material and methods:

Line 79-80: Did authors taken any criteria into account for sampling the animals? For instance, pedigree relationship, Age, weight, herd??

Line 81-83: What was the age that Body weight and body measurement information were collected? Did all animals had the same age?

A Table about the descriptive statistics of studied traits should be provided.

Lines 117-118: Did you included any fixed or random effects into the statistical model? have the association analysis conducted per breed or all breeds together?

Have you done any quality control for haplotypes and SNPs before the association analysis?

Results:

Figure 2: More clarification is needed for the color’s gradients, when the genotype frequency match with HWE test, the color is white, isn’t it? For some of loci the plot is symmetric (P3) and for some is not, does below diagonal and upper diagonal show different statistics?

Most of the Figure 2’s results are presented in Tables 3&4. I would suggest authors remove Figure 2 to avoid duplication of results.

Since the components of statistical model are not described well, it is hard to evaluate the results of association analysis. I would leave reviewing this part after more clarification about the model added to the manuscript.

Author Response

(The authors gave the same response as above.)

Reviewer 3 Report

This manuscript demonstrates a progressive approach of cattle selection based on molecular assessment of the 3’ UTR of the Prosaposin (PSAP) gene which is a known regulator of body growth in a dose-dependent manner. The authors went on to assess sequence level variation which were predicted to alter transcription factor and DNA methylating enzymes, thus potentially changing levels of transcription, however, this was not tested, in vivo. However, this manuscript does characterize four main variants found in the study population via in vitro luciferase activity assays as well as comparing overall body size parameters.

The experimental logic and approach undertaken within this manuscript was appropriate and well described using sufficient numbers of representative samples. Sanger Sequencing followed by PCR-amplification of the 3’ UTR region of the gene identified common allelic variants and those were assessed for differential luciferase activity and the resulting terminal phenotype of the animals which harbor those variants.

Broad points of clarity

  1. The use of haplotype appeared to represent a given allele and not a combination of the diploid state of PSAP gene. It is not clear to the reader whether allele combinations were assessed within the observed phenotype of the cattle. Can it be determined which allele identified was actually expressed and at what levels? Tables 5-6
  2. Figure 4 caption should be formatted to be on the same page as the figure (line 203)
  3. Interpretation of the addition of the bta-miR-184 within Luciferase activity assay, are the authors concluding that the addition of this putative PSAP-targeting miR had no effect on PSAP transcriptional regulation? It would be useful to have a positive control here to confidently interpret the effect of exogenous miR mimics within this system. What endogenous mRNA targets within the HEK293 cell line does bta-miR-184 target? Do you see any change in this transcript abundance? Without this control, this aspect of the study is uninterpretable and is suggested to be removed from the study.

Minor points

  1. Though the discussion is well-organized and cited, several lines contain correctible use of the English language:
    1. 244: excavated, multitudinous… consider different words
    2. 252-253: “effective evidences to achieve…”
    3. 254: “…get our attention”
    4. 257: …and it could be an important evidence”
    5. 270: “it was a pity that the results were not in line with our expectations”

Author Response

(The authors gave the same response as above.)
